# Genetic Insight into the Interaction of IBDV with Host—A Clue to the Development of Novel IBDV Vaccines

**DOI:** 10.3390/ijms24098255

**Published:** 2023-05-04

**Authors:** Hui Gao, Yongqiang Wang, Li Gao, Shijun J. Zheng

**Affiliations:** 1National Key Laboratory of Veterinary Public Health Security, Beijing 100193, China; gaohui121@cau.edu.cn (H.G.);; 2Key Laboratory of Animal Epidemiology of the Ministry of Agriculture, Beijing 100193, China; 3College of Veterinary Medicine, China Agricultural University, Beijing 100193, China

**Keywords:** infectious bursal disease virus (IBDV), genetic evolutionary typing, viral genome diversity, reverse genetic

## Abstract

Infectious bursal disease virus (IBDV) is an immunosuppressive pathogen causing enormous economic losses to the poultry industry across the globe. As a double-stranded RNA virus, IBDV undergoes genetic mutation or recombination in replication during circulation among flocks, leading to the generation and spread of variant or recombinant strains. In particular, the recent emergence of variant IBDV causes severe immunosuppression in chickens, affecting the efficacy of other vaccines. It seems that the genetic mutation of IBDV during the battle against host response is an effective strategy to help itself to survive. Therefore, a comprehensive understanding of the viral genome diversity will definitely help to develop effective measures for prevention and control of infectious bursal disease (IBD). In recent years, considerable progress has been made in understanding the relation of genetic mutation and genomic recombination of IBDV to its pathogenesis using the reverse genetic technique. Therefore, this review focuses on our current genetic insight into the IBDV’s genetic typing and viral genomic variation.

## 1. Introduction

Infectious bursal disease virus (IBDV) is a globally prevalent chicken immunosuppressive virus that causes severe immunosuppression in infected chickens, leading to increased susceptibility to other pathogens or even death [1,2,3]. Although there are two serotypes of IBDV, only serotypeI causes disease in poultry. SerotypeII was isolated from turkeys and is not pathogenic to chickens [3,4]. Under natural conditions, IBDV can infect all breeds of chickens, causing huge economic losses to the poultry industry worldwide [5,6,7,8,9,10].

Due to the unique bi-segmented double-stranded RNA genome and its high error rate of viral RNA-dependent RNA polymerase (RdRp) [3,11,12], IBDV is naturally prone to varying degrees of genomic mutation or recombination, leading to the emergence and spread of new mutant or recombinant strains in chickens [13,14,15]. Several main pathogenic types of IBDV have been identified, including classical IBDV (cIBDV), variant IBDV (varIBDV), very virulent IBDV (vvIBDV), attenuated IBDV, and novel variant IBDV (nVarIBDV), which differ in pathogenicity and antigenicity [7,16,17,18]. Nonetheless, there is no clear standard for typing the IBDV genome, and the description of the genotype of IBDV has become complex and imprecise as novel strains emerge [6,19,20]. Consequently, traditional classification methods based on pathogenicity and antigenicity need to be modified. Recently, several improved genotyping schemes have been proposed that would greatly facilitate the genetics and molecular epidemiological investigation of IBDV [21,22,23,24].

Currently, the vaccination of chickens with inactivated and live attenuated vaccines are commonly-used clinical methods to control IBD [25,26,27,28]. However, the high mutation rate of IBDV is likely to be the reason for the emergence of mutant viral strains whose antigenicity differs from that of the current commercially-available vaccines [16,29,30,31]. The battle between IBDV and the host is manifested in various ways [32,33,34,35], the most obvious being viral variation at a genomic level that is critical to genetic diversity and immune evasion [14,16,36,37]. Therefore, a comprehensive understanding of the patterns of genomic variations in IBDV is critical for the prevention and control of IBD. In recent years, reverse genetics has emerged as a useful technic to combat IBDV infection. This approach can make true the precise mutations or substitutions in the IBDV genome from a genetic perspective, and many advances have been made in this aspect to explore pathogenesis and develop novel vaccines [38,39,40,41].

This review focuses on the current epidemics and pathogenesis of infections with different IBDV strains, as well as novel approaches to virus classification based on phylogenetic evolution. In addition, we have summarized the current genetic insights into virus-host interactions at a genomic level, as well as the application of reverse genetics to IBDV vaccine development, providing new perspectives for future prevention, control, and vaccine development against IBDV.

## 2. Epidemics and Pathogenesis of Different IBDV Strains

### 2.1. Classical IBDV (cIBDV)

In 1957, the original outbreak of IBD occurred in the area of Gumboro, Delaware, USA, where researchers observed a high rate of disease occurrence in chicks [42]. Early symptoms of IBD include diarrhea, loss of appetite, weakness, and even death [42]. The main target organ of the virus is the bursa of Fabricius (BF) of 3–6-week-old chicks, which is characterized by enlarged or hemorrhagic bursa during the first four days, followed by bursal atrophy later in the course of the disease [3,25,43]. The infection eventually leads to lymphocyte failure and destruction of the bursa, which is the main feature of IBD pathogenesis [1,44]. In classical epidemic situations, mortality in diseased chickens may range between 1% and 50%, with significant effects on both broilers and laying hens [3]. In addition to mortality, IBDV also has an immunosuppressive effect, which affects the host's immune response and the efficacy of other vaccinations [3,43,45]. The virus genome consists of two segments of RNAase-resistant, double-stranded RNAs [46]. The complete terminal sequences of IBDV genomic dsRNA have been identified, indicating that different RNA structures may have an impact on the ability of the virus to replicate [47]. As more complete genomic information on IBDV has become available, the correlation between genetic and pathogenic phenotypes among different strains of IBDV could be more precisely assessed [14,17,48,49,50,51].

### 2.2. Variant IBDV (varIBDV)

In the late 1980s, it was first reported that IBDV variants were identified by virus neutralization tests as serotypeI IBDV with significant antigenic differences from classical strains [37,52]. These antigenically altered strains were collectively referred to as variants IBDV to distinguish them from previous classical IBDV isolates [22,42]. Initially, the variant IBDV-infected chickens were characterized by little or no mortality and no obvious clinical signs, but their bursa and spleen were damaged [29,53,54]. Subsequent studies have shown that amino acid mutations in the hypervariable region (HVR, nt 616–1050, aa 206–350) of VP2 are the major cause of IBDV variants [30,55,56]. The VP2 hypervariable region includes many amino acid residues exposed on the protein surface, and mutations in these residues can lead to a variation in the antigenicity of VP2, allowing the variant to escape from the neutralizing antibodies produced by vaccination against the cIBDV [54,56]. This phenomenon is known as antigenic drift and is the primary reason for antigenic diversity [30,57]. Several mutations in the specific residues (222 T, 249 K, 254 S, 286 I, and 318 D) in the HVR may cause antigenic drift [58,59].

### 2.3. Very Virulent IBDV (vvIBDV)

At the time the variant IBDV was identified in North America, very virulent IBDV (vvIBDV) emerged in Europe [60,61]. In contrast to the variant strains, vvIBDV typically causes high morbidity and mortality in SPF chickens, resulting in mortality of 50–100%, along with typical signs and lesions. Importantly, vvIBDV can establish infection in the presence of maternal antibodies to the classical strain and cause lesions in immune organs other than the bursa [2,25]. As a highly transmissible virus, vvIBDV has spread rapidly throughout the world, causing significant economic losses to the poultry industry [7,50,62,63,64,65]. However, the emergence of vvIBDV promoted scientific research on the pathogenicity of IBDV infection.

Genetic studies revealed that the emergence of the vvIBDV strains was due to the reassortment of genetic segments, specifically the reassortment of the mutated segment A with the segment B of unknown origin, resulting in a sudden increase in the pathogenicity of the virus [66]. Residues 222 A, 242 I, 256 I, 294 I, and 299 S of the vvIBDV segment A are conserved compared to other strains of IBDV and serve as markers of pathogenicity [2,67,68]. In addition, residues 253 and 284 of VP2 are proposed to be the determinants of cell tropism and major contributors to IBDV virulence [69,70,71]. Phylogenetic analysis has shown that segment B of vvIBDV is distinct and highly conserved [13,58,66]. Additional studies have confirmed that both genomic segments contribute to the high virulence of vvIBDV [72,73,74,75,76]. As the efforts in exploring vvIBDV continue, more critical residues that may be involved in the pathogenicity of the virus have been identified [77,78,79].

### 2.4. Novel Variant IBDV (nVarIBDV)

Since 2015, China has experienced an outbreak of the novel variant IBDV (nVarIBDV) [6]. These isolated strains have distinct subclusters, indicating genetic evolution in both segments A and B of the viral genome, which rarely occurred in China before [80,81,82,83]. While the vvIBDV strain has historically been the most prevalent with low variability in field transmission [7,17], the recent emergence of nVarIBDV outbreaks in other Asian countries such as Japan [9], South Korea [84], and Malaysia [85] are of concern. It was reported that nVarIBDV does not cause severe clinical signs but causes irreversible damage to the immune organs of chickens, including bursal lesions, spleen swelling and atrophy, and long-term immunosuppression, to a greater extent than the previous varIBDV [6,18,86]. In addition, the new strains can break through the immune protection provided by existing vvIBDV vaccines [84,87,88].

The novel variant spectrum of IBDV strains exhibits significant genetic differences from previously reported IBDV strains. The strains contain multiple amino acid substitutions, with VP2 having typical residues similar to varIBDV (222 T, 249 K, 286 I, and 318 D) [58], as well as other residues such as 221 K, 252 I, and 299 S, and VP1 containing 147 D and 508 K [6]. Although the same amino acid differences have been reported in other studies [89], the relationship between these amino acid substitutions and the antigenicity and pathogenicity of the strains has not been fully investigated. Therefore, further investigations into novel variant IBDV strains are needed to better understand the epidemics of currently circulating IBDV strains.

### 2.5. Other Strain Types of IBDV

There are several other strain types of IBDV, including attenuated, reassortant, and recombinant strains. Attenuated strains usually refer to those attenuated live vaccine strains made especially for the control of cIBDV or vvIBDV infection and are classified as mild, intermediate, or intermediate-plus based on their attenuation [25]. The immunization of chicks with attenuated vaccine strains has become the primary defense against IBD in young chickens. Although attenuated strains are generally not lethal to chickens, intermediate and intermediate plus attenuated strains can cause varying degrees of bursal damage in vaccinated chickens [26]. However, due to the widespread use of live vaccines, the spread of different strain types among flocks in recent years has led to the increasing emergence of reassortant and recombinant strains of IBDV, such as strains with segment A of vvIBDV and segment B of attenuated strains [76,90,91], segment A of vvIBDV and segment B of serotypeII [75], and segment A of vaccine strains and segment B of vvIBDV [15,92], etc. Homologous recombination between different strains of IBDV has also been reported [58,93,94,95]. Recombination and reassortment between different strains pose new challenges to the prevention and control of IBD and require continued efforts to investigate the genetics and epidemics of IBDV.

## 3. Classification of IBDV Based on Genomic Phylogenetic Evolution

### 3.1. Recent Scheme for IBDV Genomic Classification

The transmission, mutation, and recombination of IBDV between different countries and regions over the decades have led to the emergence of different genotypic strains [21,22]. Recently, as more diverse strains are discovered, some new universal genogrouping methods have been proposed [21,23,24].

Michel and Jackwood proposed a new viral classification method based on the evolutionary analysis of IBDV segment A sequences in 2017 [21]. They sequenced 90 samples from 23 countries worldwide and selected a 579 bp (nt 743–1331) fragment of segment A containing the VP2 hypervariable region for phylogenetic analysis. They classified IBDV into seven genogroups (G1–G7). And it could partially characterize the previous classification of IBDV. For example, G1 contains most classical IBDV strains, G2 contains American variants, and G3 contains vvIBDV strains. Furthermore, strains isolated from South America represent G4, while G5 and G6 include representative strains from Mexico and Italy, respectively. The Australian and Russian strains make up G7. Jackwood et al. suggested further subdivision of the different genomes into different subgroups and proposed a more specific scientific nomenclature [22]—these proposals are encouraging.

Several studies suggested that the pathogenicity of IBDV is due to both of its genomic segments and that segmental reassortment also plays an important role in viral evolution [72,96]. The genetic data of segment A alone are insufficient to characterize the potential pathogenicity of IBDV, so it is necessary to classify IBDV based on both genomic segments. Islam et al. proposed a phylogenetic analysis based on the two segments of IBDV genome and selected a 366 bp region of segment A (nt 785–1150, aa 219–340) and a 508 bp region of segment B (nt 328–835, aa 73–241), classifying the segment A of IBDV into nine genogroups and B into five (nucleotide counting starts at the 5′ terminal of the IBDV genome) [23]. The fragment selected for segment A contains a VP2 hypervariable region, and segment B contains the “B marker region” (nt 328–756, aa 110–252), which could characterize the phylogenetic evolution of segment B [97]. Thus, genogroups of segment A were classified as A0 (serotypeII), A1 (A1a: classical virulent, A1b: classical attenuated), A2 (US antigenic variant), A3 (very virulent), A4 (early European and recent South American distinct IBDV, dIBDV), A5 (atypical or recombinant Mexican strains), A6 (atypical Italian), A7 (early Australian), and A8 (Australian variant). Genogroups of segment B were classified as B1 (classical-like), B2 (very virulent-like), B3 (early Australian-like), B4 (Polish and Tanzanian), and B5 (Nigerian).

Meanwhile, Wang et al. proposed a similar scheme to classify IBDV genogroups [24], using the HVR of segment A (nt 616–1050, aa 206–350) and the B-marker of segment B (nt 328–756, aa 110–252). Similarly, IBDV is divided into nine genogroups of “A” and five of “B” in this scheme. However, the A2 of serotypeI strains is further divided into four subgroups: A2a, A2b, A2c, and A2d. The novel variant of IBDV that has recently emerged in China is classified as subgroup A2d [6]. A8 is defined as attenuated strains with specific cell tropism and non-pathogenic characteristics. B3 consisted of HLJ0504-like strains isolated in China [98], and B4 consisted mainly of recently discovered European transitional-lineage strains [99,100]. In addition, the serotypeII strains were defined as AII and BII, respectively.

All these studies have contributed greatly to the molecular epidemiology of IBDV worldwide. We have summarized their genomic classifications in Table 1 and Table 2. Despite differences in the length of genomic fragments selected for genogrouping and the philosophy behind the classifying systems, they all have similar conclusions, indicating the reliability of this system. However, there is some controversy regarding the classification of early Australian strains, Australian variants, and attenuated strains. Besides, the scheme of Wang et al. did not include the B5 (Nigerian) genogroup for analysis [101].

### 3.2. Revised Proposal for IBDV Genomic Classification

Following their principles of sequence selection [21,23,24], we revised the classification proposal by selecting a 391 bp fragment from segment A (nt 631–1021, aa 211–340) and a 528 bp fragment from segment B (nt 217–744, aa 73–248) for phylogenetic analysis of IBDV. These fragments included as much as possible the VP2 hypervariable region of segment A and the “B marker region” of segment B, as well as sequences reported in the literature and representative sequences selected by combining Islam et al. [23] and Wang et al. [24]. Sequence information for IBDV was obtained from NCBI (www.ncbi.nlm.nih.gov, accessed on 3 March 2023). The resulting phylogenetic analyses are shown in Figure 1 and Figure 2. Notably, the genomes of nVarIBDV appear as separate subgroups for both segments A and B.

Our analysis largely agrees with the classification schemes mentioned above [21,23,24], but we made some suggestions for the controversial areas. Attenuated strains are mainly derived from cIBDV or vvIBDV and could theoretically belong to the same subgroup as A1 (classical virulent) [5,26]. However, they form unique subgroups in the phylogenetic tree (Figure 1). Furthermore, the separate subgrouping of attenuated strains helps to define reassorting virulent strains. Therefore, we propose to define attenuated strains as a new subgroup called A9 (attenuated). Similar situations were observed for the early Australian and the Australian variant strains, which formed separate clades in the phylogenetic tree (Figure 1). To avoid adding further confusion to the existing genogroup classification criteria, we propose to retain the classification scheme of Islam et al. for the Australian strains as A7 (early Australian) and A8 (Australian variant). In addition, our results indicate that segment A of serotypeII has shown distinct subgroups (Figure 1), while the branches of segment B, although not forming independent subgroups (Figure 2), clearly cannot be classified as B1 (classical-like) [23]. And the existing nomenclature of either A0 [23] or AII [24] may not be clear at the level of presentation. Therefore, we propose that the genomes of serotypeII be designated as SIIA and SIIB, respectively. In addition, our phylogenetic analysis shows that the genetic lineage of segment A follows a clear stepwise pattern with distinct genogroups, where SIIA represents the most distant group (Figure 1). However, segment B exhibits more complex lineage divisions that differ from segment A (Figure 2), suggesting that the two segments may have followed different evolutionary pathways.

Overall, the phylogenetic evolution of IBDV is a complex process closely linked to genetic reorganization and the continuous emergence of new subtypes. The new genotyping scheme provides a more detailed and accurate classification of the genetic evolution of IBDV. With further improvement, it can help researchers better understand the epidemiological and molecular evolutionary mechanisms of IBDV and provide more reliable and effective theoretical support for the prevention and control of IBD.

## 4. Genetic Factors Affecting IBDV–Host Interactions

The interaction between IBDV and its host is influenced by several genetic factors, which have been well discussed in other reviews [34,103,104,105]. The variability of the IBDV genome is a critical factor affecting the interaction, involving mutations, gene reassortment and genetic recombination, and accounting for the emergence of multiple genotypes. The selection pressure associated with the widespread use of vaccines and the long-term transmission and prevalence of IBDV has resulted in the accumulation of multiple mutations and recombination in the IBDV genome sequence [7,14]. Such mutations help IBDV evade recognition and clearance by the immune system, thereby increasing its survival [30]. In this section, we focus on the amino acid mutations in IBDV that have been identified and partially studied, as well as the genome reassortment and genome recombination among the viruses.

### 4.1. Mutations in IBDV

Gene mutation is one of the major mechanisms of the genetic evolution of IBDV. Lines of evidence indicate that the VP2 hypervariable region is the most readily mutated region in the IBDV genome and has the most important impact on the antigenicity and pathogenicity of IBDV [36,56,106,107]. VP2 is the major structural protein of IBDV, serving as the primary protective antigen of the virus and inducing the production of neutralizing antibodies [108,109]. As shown in Figure 3, VP2 is folded into three structural domains, namely the base (B), the shell (S) and the projection (P) [107,110,111]. Among them, the B and S structural domains are relatively conserved, while the P domain is multivariate and contains the hypervariable region of VP2 [56,112]. Furthermore, the P domain contains four loops, namely P_BC_ (aa 204–236), P_DE_ (aa 240–265), P_FG_ (aa 270–293), and P_HI_ (aa 305–337) [56,107,110]. Previous studies have reported that the P_DE_ and P_FG_ domains mainly affect the cellular adaptability and pathogenicity of the virus [70,113,114]. In contrast, the P_BC_ and P_HI_ domains are responsible for the antigenic variability and immune escape of IBDV [30,56], which is also indicated in Table 3. Moreover, the P_BC_ and P_HI_ structural domains may also play a role in the process of virus assembly and maturation (Table 3).

Earlier studies on mutation sites in the VP2 focused on antigenic variants. For example, mutations in amino acid residues 213, 222, 223, 249, and 324 have been found to be associated with loss of responsiveness to specific monoclonal antibodies [30,53,106,112]. However, in a later study, it was found that mutations in residue 222 are involved not only in immune escape but also in viral replication and virulence [56]. Other amino acid residues, such as 286 I, 318 D, and 321 A, have also been strongly associated with antigenic variants in IBDV [53,56,59]. It was found that mutations at residues 222 and 254 can cause IBDV to break through the immunity induced by the parental Del-E strain vaccine [30]. In addition, it was recently shown that mutations in residues 318 and 323 of VP2 significantly affect the antiserum neutralization of nVarIBDV with genotype A2dB1 [117]. The results from these studies suggest that amino acid mutations located in the hypervariable region of VP2 have a crucial impact on the responsiveness of the neutralizing antibodies and can lead to successful infection of chicks by the mutant strain even when maternal antibodies remain relatively high. In addition, several amino acid residues have been found to influence the cell tropism and pathogenicity of IBDV, including D279N, which has been proven to contribute to viral adaptation to cell culture and is a marker of reduced pathogenicity [113,118]. Residues at positions 253 and 284 were shown to be the determinants affecting cell tropism, with 253 H being associated with attenuation of virulence and a main contributor to IBDV virulence [69,70,113,114,119]. Mutations in residues 249 and 256 were associated with viral replication and virulence, while the additional mutation Q249R introduced on the Q253H/A284T mutation basis in VP2 can further attenuate IBDV [77]. Mutations in residue V321A have also been related to the low pathogenicity of strain 94,432 [72]. There was some controversy regarding the role of VP2 residue 279 initially, but in a later study, it was confirmed that mutations in 279 do not contribute to IBDV virulence [71].

Furthermore, the Ile-Asp-Ala (IDA) sequence (aa 234 to 236) within the VP2 P structural domain was identified to use α4β1 integrin as a possible binding receptor for invading avian cells [78], suggesting that IBDV can use the receptor-mediated pathway to enter host cells and that VP2, as a structural protein of IBDV, must play an important role in this process. A recent study deciphered the structure of IBDV virus particles using cryo-electron microscopy and found that IBDV may use two receptors to enter cells [115]. The first receptor binds to the upper region of the P structural domain, where residues 253 and 284 determine cell tropism. The second receptor interacts with the IDA motif located in a negatively charged internalization groove, which is consistent with previous findings [78]. Moreover, residues 219 Q and 324 Q were found to contribute to the interaction between adjacent VP2 trimers and play a role in virus assembly [115].

In addition to VP2, other viral proteins of IBDV also have important roles in viral infection, invasion, and replication [105]. However, these proteins are more conserved and have been less studied in regard to gene mutations. VP1 is the RdRp of IBDV and is responsible for transcription, initial translation, and replication of the viral genome [11,120]. It was found that VP1 also contributes to IBDV pathogenicity; for example, the A276T mutation in VP1 has been shown to attenuate virulence by affecting intermolecular interactions [72]. Similarly, the V4I substitution attenuates vvIBDV virulence and increases intracellular replication [79]. In addition, the amino acid triplet 145/146/147 (TDN, TEG, or NEG) in VP1 is an important virulence site affecting the activity of RdRp, and TDN is considered a conserved marker tripeptide in vvIBDV [121]. Different structural domains of VP1 have also been shown to play separate roles in viral replication and virulence, with the N-terminal domain likely playing a more prominent role, although its exact function remains unclear [122,123]. Furthermore, it was found that nVarIBDV shares an amino acid substitution A163V with vvIBDV, which may be associated with increased pathogenicity of nVarIBDV [14]. Some recent studies have focused on post-translational modifications of VP1, demonstrating that 186 R and 426 R can be methylated by protein arginine methyltransferase (PRMT), thus affecting polymerase activity [124,125]. Likewise, studies on VP3 of IBDV have revealed that residue 235 of the VP3 C-terminus (or residue 990 of the polyprotein) can affect the replication of attenuated IBDV in vitro and in vivo. Moreover, the C-terminus of VP3 is necessary for IBDV replication [126,127].

**Table 3 ijms-24-08255-t003:** The roles of amino acid residues in IBDV mutation.

Protein	Residues	Site	Impact	Refs.
VP2	213 D	P_BC_	Immune escape	[112]
	219 Q	P_BC_	Virus assembly	[115]
	222	P_BC_	Immune escape; Virus replication and virulence-related	[30,56,59,106]
	223	P_BC_	Immune escape	[106]
	234–236 (IDA)	P_BC_	Intermolecular interactions	[78]
	249	P_DE_	Immune escape; virus replication and virulence-related	[53,77]
	253	P_DE_	Cellular adaptability; virulence-related	[69,70,114,119]
	254	P_DE_	Immune escape	[30]
	256	P_DE_	Virus replication and virulence-related	[77]
	D279N	P_FG_	Cellular adaptability	[113,118]
	284	P_FG_	Cellular adaptability	[69,70,113,114]
	286 I	P_FG_	Immune escape	[112]
	318 D	P_HI_	Immune escape	[56,59,106,117]
	321 A	P_HI_	Immune escape; virulence-related	[56,72]
	323	P_HI_	Immune escape	[117]
	324	P_HI_	Immune escape; virus assembly	[106,115]
VP1	A276T	N/A ^a^	Intermolecular interactions	[72]
	V4I	N/A	Virus replication and virulence-related	[79]
	145/146/147 (TDN ^b^, TEG or NEG)	N/A	Virus replication and virulence-related	[121]
	A163V	N/A	virulence-related(undetermined)	[14]
	R186A	N/A	Polymerase activity related	[124]
	R426A	N/A	Virus replication and polymerase activity related	[125]
VP3	235(C-terminal)	N/A	Virus replication	[126,127]

^a^ N/A—not applicable. ^b^ TDN—conserved tripeptide in the vvIBDV pathotype.

### 4.2. Gene Reassortment and Recombination in IBDV

In addition to mutations, gene reassortment and recombination are important genetic factors in IBDV that cannot be ignored. Genome reassortment of different segments in IBDV has been reported in various regions across the globe and the newly emerging variant strains are becoming a major threat to the poultry industry [100,128,129,130,131]. As the IBDV genome has subgroups of segments A and B, the widespread use of live attenuated vaccines has increased the occurrence of reassortment between vvIBDV and attenuated strains. These reassortant viruses may exhibit virulence comparable to that of classical or attenuated IBDV or may still inherit the high virulence of vvIBDV [76,132,133]. However, reassortant viruses with serotypeII segment A, regardless of the genotype of segment B, do not cause clinical disease in chickens or turkeys [129]. Therefore, evaluation of the potential virulence of reassortant viruses requires genotyping of the two genomic fragments and analysis of specific amino acid site changes.

Genome recombination is infrequent in the genetic evolution of IBDV. It is possibly due to the extreme evolutionary dynamics of segmented RNA viruses, which exhibit high rates of mutation and recombination but little homologous recombination [14]. In 2008, recombination events in IBDV segment A involving attenuated vaccine strains and two wild-type strains of vvIBDV and varIBDV were first described [93]. Subsequent studies have identified several very virulent strains whose major putative parents are vaccine strains but whose hypervariable regions are from vvIBDV, with recombination breakpoints mainly at 636 nt and 1743 nt [94]. Recently, homologous recombination was also found to occur between a nVarIBDV and an intermediate vaccine strain, resulting in increased pathogenicity of the nVarIBDV strain to chicken embryos, with recombination breakpoints at 1538 nt [134]. Another study has shown that a field isolate underwent both reassortment and recombination, resulting in enhanced virulence of the intermediate vaccine strain, with recombination breakpoints at 1468 nt and 1648 nt [95]. These events suggest that genetic recombination could occur naturally between different strains and plays a role in IBDV genetic diversity. Interestingly, almost all of the recombination breakpoints identified to date have occurred at either end of the VP2 hypervariable regions, which appear to have a unique propensity for recombination. The role of these regions in the genetic evolution of IBDV requires further investigation.

As the genetic factors affecting IBDV–host interactions are complex and diverse, mutations, reassortment, and recombination are among the important factors influencing the interaction. This variability of IBDV may affect its infectivity and virulence for host cells, as well as its ability for immune escape. Further studies on the genetic evolution and genotyping of IBDV are crucial for a better understanding of the pathogenesis of IBDV infection.

## 5. Role of Reverse Genetics in Research on IBDV Genomic Function and Vaccine Development

Reverse genetics is a powerful tool for studying the function of viral genomes. Since the successful construction and rescue of the IBDV infectious clone [135,136], significant progress has been made in understanding the relationship between the genetic variation of IBDV and its pathogenesis [137,138]. Most of the aforementioned findings have been made using reverse genetic techniques. In addition to exploring the function of the viral genome, reverse genetic techniques are a promising tool for the development of vaccines.

Several excellent reviews have covered the development of vaccines against IBDV [5,26,105]. Although various types of vaccines have been developed for IBDV, there is still a large demand for novel effective vaccines. Emerging variant strains can overcome maternal immunity induced by commercial vaccines, indicating that current vaccines are not suitable for the control of the epidemics caused by such strains [9,31,100,129,139]. The reverse genetic technique provides a new tool for vaccine development compared to traditional inactivated and attenuated vaccines. To date, reverse genetic technique has been employed to generate different rescued strains with great potential for vaccine candidates. By inserting the VP2 sequence from circulating strains into the backbone of vaccine strains, several chimeric viruses have been generated, and vaccination of chickens with the chimeric viruses could effectively protect flocks against parental strains [39,140,141]. Knocking out VP5 of IBDV produced a VP5-deficient strain, and chickens immunized with this strain were resistant to challenges with the parental virus [138,142]. Attenuated IBDV, produced by reducing the RNA polymerase activity of VP1, can induce immune protection [143]. In addition, the recombinant viruses generated by introducing the Q253H/A284T mutation into VP2 of the endemic strain and replacing it in the backbone of the attenuated strain could confer immune protection against nVarIBDV [88]. As novel mutant strains of IBDV continue to emerge, the current strategies for the prevention and control of IBD encounter new challenges. The reverse genetic techniques provide an effective approach to the development of novel IBDV vaccines that may hold great promise for the prevention and control of IBD.

## 6. Conclusions

Investigation into the pathogenesis and immune control of IBDV has been ongoing for decades. Although excellent progress has been made, frequent occurrences of IBD serve as a reminder that prevention and control methods for IBD need to be further explored. As studies of the IBDV genome and protein function progress, several questions need to be addressed. For example, what are the consequences of the involvement of genomic changes in the IBDV–host interaction? Can reverse genetic techniques be used to develop an optimal live vaccine that can provide full protection against IBD without causing damage to the bursa of Fabricius? In addition, what is the exact mechanism underlying amino acid mutations that alter the virulence of IBDV? Of note, as different chicken breeds have varying susceptibility to IBDV [32], anti-defense breeding approaches are also worth exploring to combat IBDV infection. Although live attenuated vaccines have been routinely used for the clinical control of IBD, the rapid generation of attenuated vaccine strains by reverse genetic techniques is definitely a promising option to combat the variant strains. However, potential threats, such as reassortment between vaccine and endemic strains and reversion of live vaccine virulence, must be carefully considered in vaccine development. Considering that the elucidation of the genomic diversity and variability of IBDV is crucial for understanding viral evolution, antigenicity, pathogenicity, and vaccine development, more efforts will be required to delve deeper into the mechanisms of IBDV–host interactions to provide necessary clues for the manipulation of IBDV by reverse genetics, which ultimately lead to the development of novel effective vaccines for prevention and control of IBD outbreaks.

## Figures and Tables

**Figure 1 ijms-24-08255-f001:**
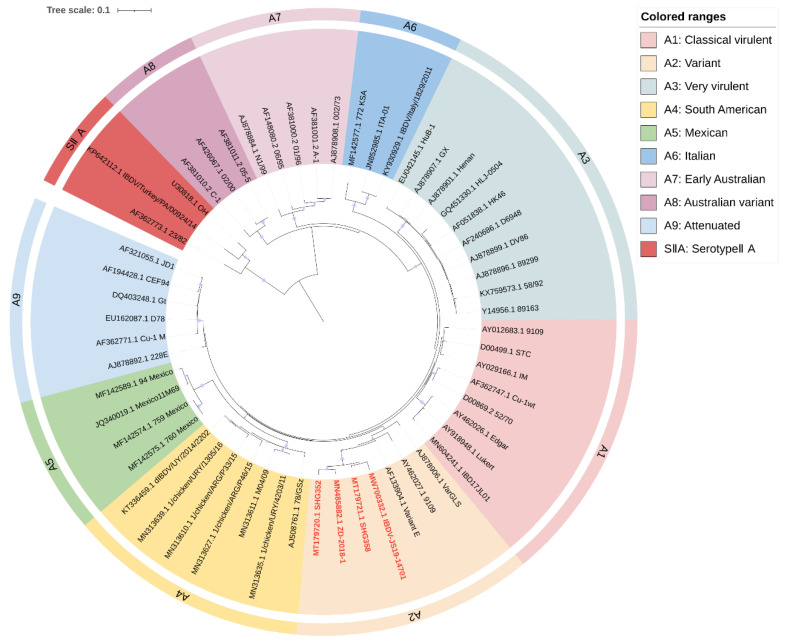
Concise circular phylogenetic analysis based on the VP2 hypervariable region of IBDV segment A. The analysis of 56 IBDV strains was performed using MEGA X [102] with the maximum likelihood method, and 1000 bootstrap replications were included. The tree was annotated in iTOL (https://itol.embl.de, accessed on 3 March 2023) and drawn to scale, with genogroup information displayed in colored circular stripes on the outermost ring. The names and GenBank accession numbers of the strains are shown in the figure. The novel variant strains recently reported belonging to the A2 are shown in red bold.

**Figure 2 ijms-24-08255-f002:**
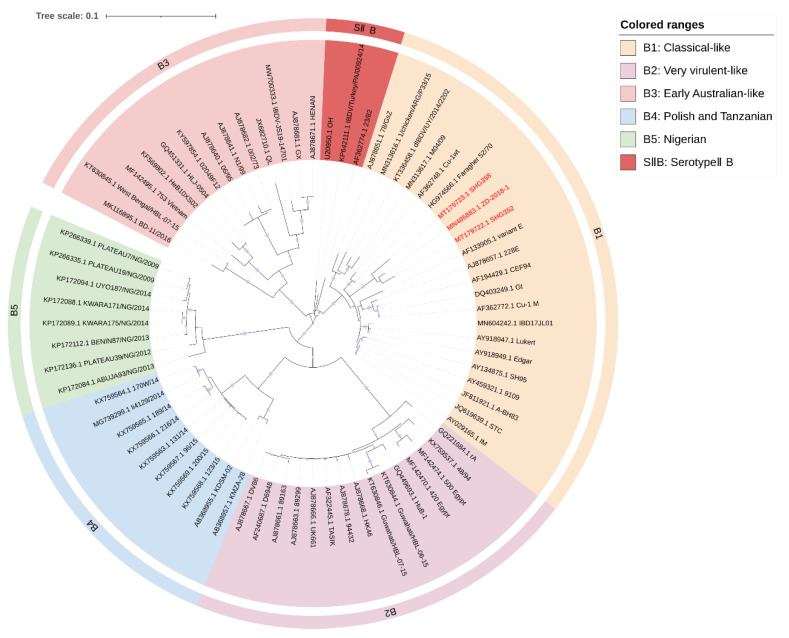
Concise circular phylogenetic analysis based on the “B marker region” of IBDV segment B. The analysis of 71 IBDV strains was performed using MEGA X [102] with the maximum likelihood method, and 1000 bootstrap replications were included. The tree was annotated in iTOL (https://itol.embl.de, accessed on 3 March 2023) and drawn to scale, with genogroup information displayed in colored circular stripes on the outermost ring. The names and GenBank accession numbers of the strains are shown in the figure. The novel variant strains recently reported belonging to the B1 are shown in red bold.

**Figure 3 ijms-24-08255-f003:**
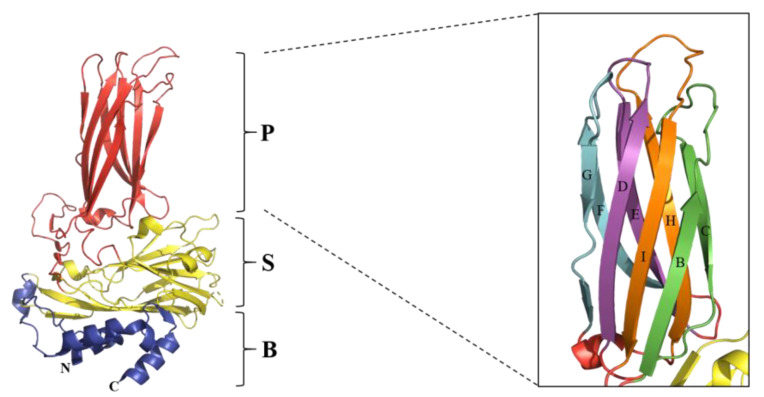
Ribbon diagram of the predictive structure of IBDV VP2. The Protein Data Bank accession code is PDB-7VRP [115]. The VP2 subunit is folded into three domains: P domain (red), S domain (yellow), and B domain (blue). The right side of the figure shows the β-sheets of the P domain, with the four loops marked in different colors and the names in uppercase letters. The figure was generated using PyMOL (The PyMOL Molecular Graphics System) [116].

**Table 1 ijms-24-08255-t001:** Genogroups of IBDV segment A.

Genetic Characteristics	Genogroups by the Following Authors			
	Michel and Jackwood [21]	Islam et al. [23]	Wang et al. [24]	Gao et al.
Classical (Virulent)	G1	A1a	A1	A1
Variant	G2	A2	A2	A2
Very virulent	G3	A3	A3	A3
South American	G4	A4	A4	A4
Mexican	G5	A5	A5	A5
Italian	G6	A6	A6	A6
Early Australian	G7	A7	A7	A7
Australian variant		A8		A8
Attenuated	N/A ^a^	A1b	A8	A9
SerotypeIIA	N/A	A0	AII	SIIA

^a^ N/A, not applicable.

**Table 2 ijms-24-08255-t002:** Genogroups of IBDV segment B.

Genetic Characteristics	Genogroups by the Following Authors			
	Michel and Jackwood [21]	Islam et al. [23]	Wang et al. [24]	Gao et al.
Classical-like	N/A ^a^	B1	B1	B1
Very virulent-like	N/A	B2	B2	B2
Early Australian-like	N/A	B3	B3	B3
Polish and Tanzanian	N/A	B4	B4	B4
Nigerian	N/A	B5	N/A	B5
SerotypeIIB	N/A	B1	BII	SIIB

^a^ N/A, not applicable.

## Data Availability

Data available on request from the authors.

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
