# Peer review of "Genetic Insight into the Interaction of IBDV with Host—A Clue to the Development of Novel IBDV Vaccines"

_ijms, 2023, doi:10.3390/ijms24098255_

Round 1

Reviewer 1 Report

The manuscript provides a thorough review on our current understanding of IBDV, focusing on genetic factors associated with virus-host interactions.

The general introduction chapter is followed by a historical overview of IBD in a well-structured format. Chapter 3 presents the latest genetic classification systems available to analyze IBDV. The proposed A9 genogroup can be a useful addition to the classification scheme, hopefully the scientific community will reach a consensus on the IBDV genotyping system. This will allow a better tool for future research on the virus. The article emphasizes the importance of analyzing both segments A and B to achieve a better understanding. It might be useful to mention here, that the characteristics of given strains are determined by the combination of specific A and B segments. Figure 1 shows that genetic lineage of segment A follows a clear stepwise pattern where genogroups appear distinct from each other, SiiA being the most distant. However, in case of segment B the lineage divisions are more complicated and different from segment A, indicating the to segments follow different evolutionary pathways. Chapter 4 summarizes the known genetic factors, associated with host interactions, as virulence factors, attenuation markers, tissue culture adaption related mutations. The authors highlight the importance of gene reassortment on the recent evolutionary developments of IBDV. Chapter 5 presents that the accumulated knowledge on IBDV genetics can be applied for vaccine developmental purposes.

The references are comprehensive (142 total), well supporting the agenda in each chapter and allow readers to find relevant research papers, to each topic presented in the article.

Overall, I would like to recommend the reviewed article to anyone interested in IBDV in general, the clear and comprehensive structure is a great help to understand our current picture on the virus.

Reviewer 2 Report

This manuscript reviewed current IBDV strains, IBDV genotype classification, genetic insights into virus-host interactions at a genomic level, and reverse vaccine, which provided new perspectives for the prevention and control of IBDV. Here are some suggestions.

1. The vv in vvIBDV means very virulent. To avoid confusion, as reference 7, it is better to abbreviate variant IBDV as varIBDV, and abbreviate novel variant IBDV as nVarIBDV.

2. In line 191-192, according to reference 24, it should be “using the HVR of segment A (nt 616-1050, aa 206-350) and the B-marker of segment B (nt 328-756, aa 110-252)”.

3. In line 214-215, could you briefly describe the reasons or reference for selecting these representative fragment.

4. Line 262, “the interaction of IBDV with the host”, the word 'with' might be not used properly.

5. About reverse genetics of IBDV, a reference of “An improved method for infectious bursal disease virus rescue using RNA polymerase II system. J Virol Methods, 2007, 142: 81-88” can be cited.
